# Game Analysis of Wind Storage Joint Ventures Participation in Power Market Based on a Double-Layer Stochastic Optimization Model

**Bin Ma** [1,2], **Shiping Geng** [1], **Caixia Tan** [1,*], **Dongxiao Niu** [1] and **Zhijin He** [1]

[1]  North China Electric Power University, Beijing 102206, China; playbestbeijing@126.com (B.M.); GengSP@ncepu.edu.cn (S.G.); Niudx@126.com (D.N.); hezhijin13579@126.com (Z.H.)
[2]  North China Power Engineering Co. Ltd. of China Power Engineering Consulting Group, Beijing 100120, China
*  Correspondence: cx_sp@ncepu.edu.cn

**Abstract:** The volatility of a new energy output leads to bidding bias when participating in the power market competition. A pumped storage power station is an ideal method of stabilizing new energy volatility. Therefore, wind power suppliers and pumped storage power stations first form wind storage joint ventures to participate in power market competition. At the same time, middlemen are introduced, constructing an upper-level game model (considering power producers and wind storage joint ventures) that forms equilibrium results of bidding competition in the wholesale and power distribution markets. Based on the equilibrium result of the upper-level model, a lower model is constructed to distribute the profits from wind storage joint ventures. The profits of each wind storage joint venture, wind power supplier, and pumped storage power station are obtained by the Nash negotiation and the Shapely value method. Finally, a case study is conducted. The results show that the wind storage joint ventures can improve the economics of the system. Further, the middlemen can smooth the rapid fluctuation of power price in the distribution and wholesale market, maintaining a smooth and efficient operation of the electricity market. These findings provide information for the design of an electricity market competition mechanism and the promotion of new energy power generation.

**Keywords:** wind storage joint ventures; power market; Nash negotiation; Shapely value

## 1. Introduction

With the deterioration of the environment and the frequent occurrence of energy problems, new energy research is being actively conducted in various fields to achieve sustainable energy development, and distributed power sources are combined with the power grid to improve the energy structure [1,2]. Distributed power generation has the characteristics of volatility, randomness, and a small amount of power trading; therefore, it does not have the conditions to participate in the competition in wholesale, futures, and contract markets. However, it can participate in the distribution market's bidding competition [3,4], changing the interests of the internal distribution network, and having an impact on the competition mechanism of the power market. Therefore, it is important to study how distributed power generation can participate in the power market competition, and the game between various stakeholders [5].

The random nature of wind power output leads to a deviation between the winning bid and the actual output. Therefore, wind power suppliers must adopt relevant technology to improve the coincidence between the actual output and the winning bid [6,7]. Common technologies include energy

storage batteries, electric gas conversion, heat storage systems, and pumped storage. Among them, pumped storage power stations are currently one of the most suitable technologies, because they can stabilize the fluctuation of wind power output and adjust the bidding bias of wind power [8–10]. This paper investigates wind power suppliers and pumped storage power stations as joint wind storage ventures to participate in the bidding competition of the electricity market, distributing the profit of cooperative games between wind power suppliers and pumped storage power stations.

A substantial amount of research has been conducted regarding the construction and application of wind storage participation in the electricity market equilibrium model. In terms of model construction, each researcher builds on different theories. Among them, Ding, Tomasz et al. [11,12] proposes a wind storage joint operation model based on the independent operation model. Based on the wind power random output model, Wang, Zhao et al. [13] proposed a joint strategy of wind storage. In terms of model application, Li, Liu et al. [14] combined wind storage with the linkage game problem of the electricity market. Diaz, Coto et al. [15] provided a comparative analysis of the differences between the wind energy quotient and the energy storage quotient under two models of joint venture concluded that the wind storage joint venture can increase the income of both and is applied to multi-stage market bidding. Based on the combination of wind storage, comprehensive consideration of wind power's uncertain output, and dynamic changes in bidding market prices, the optimal bidding sequence is proposed. The above research focuses on the effect of wind storage coalitions on wind power fluctuations, the income of wind power quotients, and energy storage quotients, but rarely studies the impact of wind storage on the cost-effectiveness of other market entities and the income distribution of participating in competition of the power distribution market.

With regard to the study of profit distribution in joint games, researchers have proposed different methods. [16,17] proposed the Shapely value to distribute the income, but the application scenarios were different, Wu, Zhou et al. [16] based on the Shapley value to share the income between wind power suppliers and pumped storage power stations. Yang, Tan et al. [17] constructed an alliance operation optimization model for power producers, and used the improved Shapely algorithm to allocate the net income of power producers. Zhang, Zhang et al. [18] used the Nash negotiation method to obtain the income plan of the incremental distribution network. In addition to using a single allocation method, Liu, Chu et al. [19] mixed multiple distribution methods using the nucleolar model and the Shapley value distribution method to achieve the income distribution of each control area in the AGC coordinated control.

Compared with the existing research, the innovations of this paper are as follows: (1) Through the aggregation of distributed wind power and pumped storage power stations, it is regarded as a deterministic power supply, participating in the power distribution market's bidding competition, and stabilizing the volatility of wind power generation. (2) The introduction of middlemen in the wholesale and distribution market to realize power trading. (3) The Nash negotiation method and Shapely value are used to quantify the profit between the wind storage joint venture, the wind power suppliers, and the pumped storage power stations, which solves the problem of cooperative game profit distribution.

The remainder of this paper is organized as follows: In Section 2, we construct the upper-level game model of power producers and wind storage joint ventures, establish a market trading framework, propose model assumptions, and establish a decision-maker model for power producers, middlemen, and wind storage joint ventures. In Section 3, based on the wind storage joint venture profit to construct the lower-level distribution model, the Nash negotiation method is used to distribute the profit between the wind storage joint ventures. The Shapely value is used to distribute the profit between the wind power suppliers and the pumped storage power stations. In Section 4, the model calculation method and solution are processed. In Section 5, the correctness of the two-layer stochastic optimization model is verified by a case study, which provides a reference for distributed power generation to participate in the bidding competition of the power market. Section 6 highlights the main conclusions of the paper.

## 2. Upper-Level Power Producer-Wind Storage Joint Venture Game Model

### 2.1. Market Trading Framework

Based on the trading framework of wind storage joint ventures participating in the electricity market (Figure 1), in the wholesale market, a total of *G* power producers participate in the tender to determine the power price of the wholesale market. In the distribution market, wind power and pumped storage power stations form *N* wind storage joint ventures to participate in the bidding, determining the power price of the distribution market. Among them, the operation models of the wind storage joint ventures are adjusted to achieve the maximum benefit in real time according to the actual output and the winning bid. When the actual output is greater than the winning bid, the bid deviation is positive, and the pumped storage power station stores electricity. When the actual output is less than the winning bid, the bid deviation is negative, and the pumped storage power station generates electricity. Finally, the profit is distributed based on the Nash negotiation method and the Shapely value. The middlemen coordinate to realize the power transaction between the wholesale and distribution markets. When the price of the wholesale is greater than the price of the distribution market, the middlemen participate in the distribution market bidding competition, and the wholesale market sells the winning electricity in the wholesale market. When the market power price is greater than that of the wholesale market, the middlemen participate in the bidding competition of the wholesale market and sell the winning electricity in the distribution market.

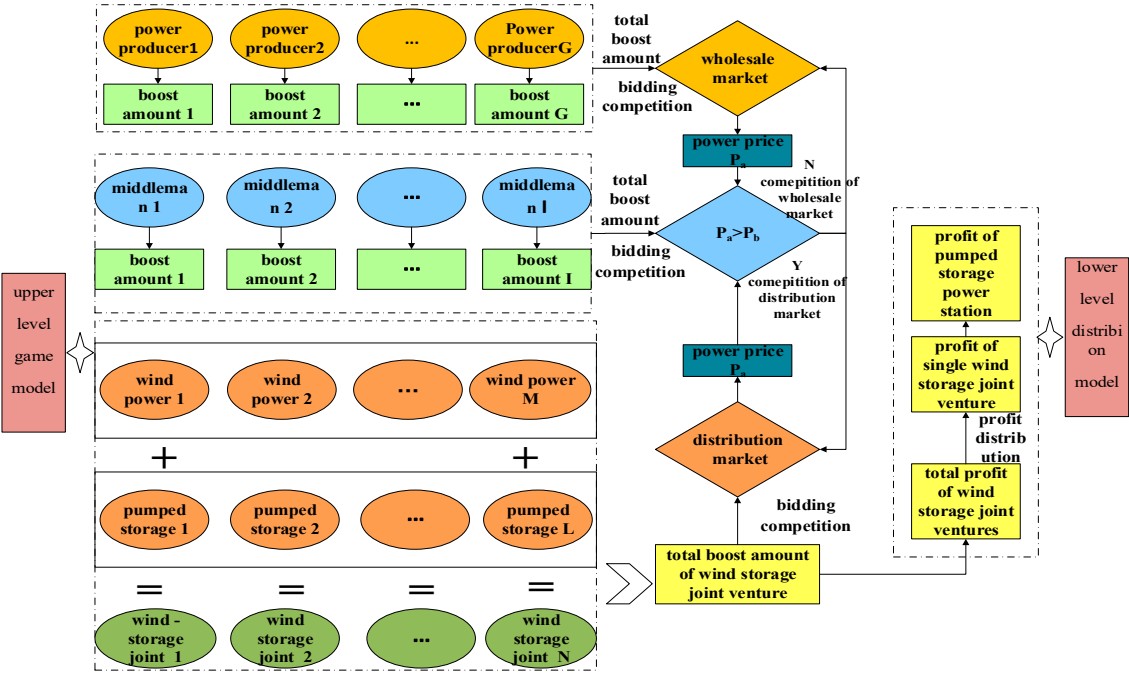

**Figure 1.** Wind storage joint venture to participate in the trading framework of the power market.

(1) The bid amount of the power producer, middleman, and wind storage joint ventures is equal to the boost amount, and the deviation between the bid amount and the boost amount is not considered.

(2) The capacity of the pumped storage power station is large, and it can flexibly adjust the deviation between the actual output and the winning output. It can completely suppress the uncertainty of wind power output and consider the wind storage joint ventures as a definite output. However, within the wind storage joint venture, wind power output is uncertain.

The upper-level game model involves the two major markets of wholesale and distribution, the three main players of power producers, wind storage joint ventures, and middlemen. The three

main players constantly adjust the game plan according to the decision-making model in the two major markets, and finally achieve equilibrium of the two markets. The three main game scenarios are as follows.

*2.2. Decision Model of Power Producers*

As the main competition players of the wholesale market, the power producers include large-scale conventional energy generators. Each power producer participates in the market competition through its own supply function. Finally, the decision-making behavior of each power producer determines the price of the wholesale market.

2.2.1. Objective Function

Assuming that a total of G power producers participate in the competition of the wholesale market (with the aim of maximizing profits), and accordingly establishing the following power generation decision model, the objective function can be expressed as follows:

$$maxR_{con,g} = P_a Q_{con,g} - C(Q_{con,g}) \tag{1}$$

where, $R_{con,g}$ represents the profit of power producer $g$; $P_a$ represents the power price determined by wholesale market; $Q_{con,g}$ represents the bidder of the power producer; $C(Q_{con,g})$ represents the cost of power producer $g$.

Cost function of the generator $C_{con,g}$, supply function $Q_{con,g}$, it can be further expressed as follows:

$$C(Q_{con,g}) = a_{con,g} Q_{con,g}^2 + b_{con,g} Q_{con,g} + c_{con,g} \tag{2}$$

$$P_a = \beta_{con,g}(b_{con,g} + a_{con,g} Q_{con,g}) \tag{3}$$

where, $a_{con,g}$ and $b_{con,g}$ represents the variable cost factor of the generator $g$; $c_{con,g}$ represents fixed cost factor for the generator $g$; $\beta_{con,g}$ represents the bidding strategy parameter of the generator $g$ which equals the ratio of the generator's quotation to the marginal cost. The larger the ratio, the higher the quotation of the generator.

2.2.2. Restrictions

The generator constraints include supply and demand balance constraints, and upper and lower constraints of variables. The balance of supply and demand is the total power generation being equal to the total market demand. The upper and lower limits of the variable are the non-negative cost coefficient of the generator and the upper and lower limits of the bid strategy parameters. This is detailed as follows:

$$\sum_{g=1}^{G} Q_{con,g} + Q_{con,dis} = Q_{con,sale} + D_a \tag{4}$$

$$a_{con,g}, b_{con,g}, c_{con,g} \geq 0 \tag{5}$$

$$\beta_{\min} \leq \beta_{con,g} \leq \beta_{\max} \tag{6}$$

where, $Q_{con,sale}$ represents the bid amount of the middlemen in the wholesale market; $Q_{con,dis}$ represents the bid amount of the middleman in the distribution market; $D_a$ represents the demand for the wholesale market.

*2.3. Decision Model of Wind Storage Joint Venture*

As a competition entity in the distribution market, wind storage joint ventures include small wind power producers and large-capacity pumped storage power stations. Because of the randomness of the wind power output, pumped storage power plants can first be flexibly adjusted according to their

output deviation, and second (to participate in market competition) according to the overall supply function of the wind storage joint venture. Finally, the decision-making behavior determines the power price of the distribution market.

### 2.3.1. Objective Function

It is assumed that a number of wind power suppliers and pumped storage power stations form $N$ wind storage joint ventures, aiming at maximizing the overall profit of wind storage joint ventures, and establishing the following decision model of wind storage joint venture. The objective function of the wind storage joint ventures can be expressed as follows:

$$maxR_{wind\text{-}ESS,n} = P_b Q_{wind\text{-}ESS,n} - C(Q_{wind\text{-}ESS,n}) \tag{7}$$

where, $R_{wind\text{-}ESS,n}$ represents the joint profit for wind storage joint venture $n$; $P_b$ represents the power price determined by the distribution market; $Q_{wind\text{-}ESS,n}$ represents the boost amount of the wind storage joint venture $n$; $C(Q_{wind\text{-}ESS,n})$ represents the cost for wind storage joint venture $n$.

Cost function of the wind storage joint venture $C_{wind\text{-}ESS,n}$; supply function $Q_{wind\text{-}ESS,n}$ can be further expressed as follows:

$$C(Q_{wind\text{-}ESS,n}) = a_{wind\text{-}ESS,n}Q^2_{wind\text{-}ESS} + b_{wind\text{-}ESS,n}Q_{wind\text{-}ESS,n} + c_{wind\text{-}ESS,n} \tag{8}$$

$$P_b = \beta_{wind\text{-}ESS,n}(b_{wind\text{-}ESS,n} + a_{wind\text{-}ESS,n}Q_{wind\text{-}ESS,n}) \tag{9}$$

where, $a_{wind\text{-}ESS,n}$ and $b_{wind\text{-}ESS,n}$ represent the variable cost factors of the wind storage joint venture $n$; $c_{wind\text{-}ESS,n}$ represents fixed cost factor for the wind storage joint venture $n$; $\beta_{wind\text{-}ESS,n}$ represents the bidding strategy parameter of the wind storage joint venture $n$ which equals the ratio of the wind storage joint venture's quotation to the marginal cost. The larger the ratio, the higher the quotation of the wind storage joint venture.

### 2.3.2. Restrictions

The constraints of the wind storage joint venture include the overall constraints of the wind storage joint venture and the separate constraints of the wind power supplier and the pumped storage power station.

(1) Overall constraints

The overall constraints include supply and demand balance constraints and the upper and lower bounds of variables. The supply and demand balance constraint is the total power generation in the distribution market being equal to the total market demand. The upper and lower limits of the variable are the non-negative cost coefficient of the wind storage joint venture and the upper and lower limits of the bid strategy parameters. This is detailed as follows:

$$\sum_{n=1}^{N} Q_{wind\text{-}ESS,n} + Q_{con,sale} = Q_{con,dis} + D_b \tag{10}$$

$$a_{win\text{-}ESS,n}, b_{wind\text{-}ESS,n}, c_{wind\text{-}ESS,n} \geq 0 \tag{11}$$

$$\beta_{\min} \leq \beta_{win\text{-}ESS,n} \leq \beta_{\max} \tag{12}$$

where, $D_b$ represents the demand for the distribution market.

(2) Wind power supplier constraints

Assuming that a total of *M* wind power suppliers and *L* pumped storage power stations are participating in the distribution market competition, wind power supplier constraints include power balance constraints and output limit constraints, as shown in the following Equations (13) and (14):

$$\sum_{m=1}^{M} Q_{wind,m} + \sum_{l=1}^{L} Q_{ESS,l}^{dis} - \sum_{l=1}^{L} Q_{ESS,l}^{ch} - \sum_{n=1}^{N} Q_{wind\text{-}ESS,n} = 0 \tag{13}$$

$$0 \le Q_{wind\text{-}ESS,m} \le Q_{\max} \tag{14}$$

where, $Q_{wind,m}$ represents the actual output value of wind power supplier *m*; $Q_{ESS,l}^{dis}$ represents the output value of pumping unit; $Q_{ESS,l}^{ch}$ represents the pumping output value of pumping unit; $Q_{\max}$ represents the upper limit of wind power supplier's output.

(3) Pumped storage power station constraints

Pumped storage power stations include storage capacity constraints and power generation pumping limit constraints. The storage capacity constraints are shown in Equations (15)–(18):

$$S_t^{upper} = S_{t-1}^{upper} + \lambda \sum_{l=1}^{L} Q_{ESS,l}^{ch} - \sum_{l=1}^{L} Q_{ESS,l}^{dis} \tag{15}$$

$$S_t^{lower} = S_{t-1}^{lower} + \sum_{l=1}^{L} Q_{ESS,l}^{dis} - \lambda \sum_{l=1}^{L} Q_{ESS,l}^{ch} \tag{16}$$

$$S_{\min}^{upper} \le S_t^{upper} \le S_{\max}^{upper} \tag{17}$$

$$S_{\min}^{lower} \le S_t^{lower} \le S_{\max}^{lower} \tag{18}$$

where, $S_t^{upper}$ represents the water storage at time *t* of the upper reservoir; $\lambda$ represents the operating efficiency of the pumped storage units; $S_t^{lower}$ represents the water storage at time *t* of the lower reservoir; $S_{\min}^{upper}$ and $S_{\max}^{upper}$ represent the minimum and maximum storage capacity values of the upper reservoir; $S_{\min}^{lower}$ and $S_{\max}^{lower}$ represent the minimum and maximum storage capacity values of the lower reservoir.

The power generation pumping limit constraint is shown in Equations (19)–(21). Equation (21) indicates that it is impossible for the pumping unit to be in both the pumping and generating states.

$$Q_{\min}^{ch} f \le \sum_{l=1}^{L} Q_{ESS,l}^{ch} \le Q_{\max}^{ch} F \tag{19}$$

$$0 \le \sum_{l=1}^{L} Q_{ESS,l}^{dis} \le Q_{\max}^{dis} \tag{20}$$

$$Q_{ESS,l}^{dis} \cdot Q_{ESS,l}^{ch} = 0 \tag{21}$$

where, $Q_{\max}^{ch}$ and $Q_{\min}^{ch}$ represent the output's upper and lower limits of the pumping unit when pumping; *f* represents the number of pumping units that started when pumping; *F* represents the total number of pumping units; $Q_{\max}^{dis}$ represents the upper limit of power generation for the pumping units.

*2.4. Decision Model of the Middleman*

The middleman as a transaction connects the distribution market with the wholesale market, which uses the power price difference between the wholesale and the distribution market to continuously adjust the decision-making behavior and obtain profits through the spread.

2.4.1. Objective Function

The middlemen adjust their strategies according to the power prices of the wholesale and distribution market to maximize the profits. Based on this, the following middleman decision model is established:

$$maxR_{middle} = \begin{cases} (P_a - P_b)Q_{con,sale} & P_a \geq P_b \\ (P_b - P_a)Q_{con,dis} & P_b < P_a \end{cases} \tag{22}$$

$$Q_{con,sale} = \theta_a P_a \tag{23}$$

$$Q_{con,dis} = \theta_b P_b \tag{24}$$

where, $R_{middle}$ represents the profit obtained by the middlemen; $\theta_a$ represents middlemen's determining behavioral variables in the wholesale market; $\theta_b$ represents the middlemen's determining behavioral variables in the distribution market.

2.4.2. Restrictions

Middlemen's constraints include the supply and demand balance constraints of the distribution and wholesale market, and the upper and lower constraints of the decision variables. The details of this are as follows:

$$0 \leq \theta_a \leq \theta_a^{\max} \tag{25}$$

$$0 \leq \theta_b \leq \theta_b^{\max} \tag{26}$$

$$\sum_{g=1}^{G} Q_{con,g} + Q_{con,dis} = D_a + Q_{con,sale} \tag{27}$$

$$\sum_{n=1}^{N} Q_{wind-ESS,n} + Q_{con,sale} = D_b + Q_{con,dis} \tag{28}$$

In summary, the equilibrium solution of the upper-level game model can obtain the profits of the three main players but the wind storage joint venture includes wind power companies and pumped storage power stations. Therefore, it is necessary to allocate the overall profit of the wind storage joint venture in the lower model, obtaining the profits of each wind storage joint venture, wind power supplier, and pumped storage power station.

## 3. Lower-Level Wind Storage Joint Venture Profit Distribution Model

The lower-level distribution model is mainly used to solve the problem of profit distribution among various wind storage joint ventures and internal wind storage joint ventures. Based on the satisfaction of all wind storage joint ventures, the Nash negotiation method is used to distribute profits among various wind storage joint ventures, achieving high satisfaction of all participants. According to the profits distributed by the various wind storage joint ventures, Tan, Li et al. [20] proposed that the advantage of the Shapely value is easy to understand and the feasibility is high. The profit is distributed among the internal wind storage joint ventures, which is between the wind power supplier and the pumped storage power station.

*3.1. Profit Distribution Model of Wind Storage Joint Venture Based on Nash Negotiation Method*

Based on the Nash negotiation method, an allocation scheme is proposed by each participating entity to form an allocation scheme matrix. According to the distribution plan matrix, the overall satisfaction and the most unsatisfactory plan are obtained. Finally, the profit of each participant is obtained. The specific model is as follows:

A total of $N$ wind storage joints venture in the upper model, and each venture proposes a distribution plan. Assume that the scheme $F_e$ proposed by the wind storage joints venture $e$:

$$F_e = (f_{1e}, f_{2e}, f_{3e}, \ldots, f_{Ne}) \tag{29}$$

$$\sum_{j=1}^{N} f_{je} = 1 \tag{30}$$

$$0 \leq f_{je} \leq 1 \tag{31}$$

where, $f_{je}$ represents the distribution ratio of joint venture $j$ considered by the wind storage joint venture $e$.

According to the distribution ratio of the joint venture, the allocation matrix of wind storage joint venture $N$ can be obtained.

$$F = \begin{bmatrix} f_{11} & f_{21} & \cdots & f_{N1} \\ f_{12} & f_{22} & \cdots & f_{N2} \\ \vdots & \vdots & \vdots & \vdots \\ f_{1N} & f_{2N} & \cdots & f_{NN} \end{bmatrix} \tag{32}$$

In this negotiation, the highest distribution profit ratio of the wind storage joint venture $e$ is $f_e^{\max} = \max\{ \begin{matrix} f_{e1} & f_{e2} & \cdots & f_{eN} \end{matrix} \}$; the minimum distribution profit ratio is $f_e^{\min} = \max\{ \begin{matrix} f_{e1} & f_{e2} & \cdots & f_{eN} \end{matrix} \}$. Therefore, the overall most satisfactory distribution plan is $f^{\max} = \{ \begin{matrix} f_1^{\max} & f_2^{\max} & \cdots & f_N^{\max} \end{matrix} \}$; the most unsatisfactory scenario is $f^{\min} = \{ \begin{matrix} f_1^{\min} & f_2^{\min} & \cdots & f_N^{\min} \end{matrix} \}$. If the most satisfied and least satisfied schemes are selected, the constraint of Equation (30) is not satisfied, so the adjustment coefficient $z_e$ is introduced, which meets the constraints of Equation (30) and makes wind storage joints ventures $N$ satisfied.

$$f_e = f_e^{\max} - z_e \tag{33}$$

$$R_e = f_e R_{2n} \tag{34}$$

$$f_e > f_e^{\min} \tag{35}$$

where, $f_e$ represents the profit distribution ratio of wind storage joint venture $e$; $R_e$ represents the wind storage joint venture $e$ distributed profits. Equation (35) shows that the profit distribution's proportion of each wind storage joint venture should be greater than the profit of the minimum, otherwise the negotiation fails.

*3.2. Profit Distribution Model of Wind Power and Pumped Storage Power Station Based on Shapely Value*

The wind-storage joint venture profit distribution strategy based on the Shapely value is used to solve the profit distribution between wind power suppliers and pumped storage power stations. The profit distribution plan is determined by the probability of forming a cooperative alliance and the contribution of the cooperative members.

To set up a joint game, the wind power supplier and the pumped storage power station are set to $M$. $m$ is the total number of participating members in the set $M$; $T$ is a wind storage joint venture formed by set $M$; $t$ is the number of members in $T$. The probability that a wind power supplier and a pumped storage power station randomly form a joint venture is shown in Equation (36):

$$p = \frac{1}{m!} \tag{36}$$

The probability of a specific wind storage joint venture with the cooperation scale number $t$ is shown in Equation (37):

$$p_1 = p(t-1)!(m-t)! = \frac{(t-1)!(m-t)!}{m!} \tag{37}$$

The contribution of the cooperative member $i$ in the wind storage joint venture is as shown in Equation (38):

$$\omega_i = v_t - v_{t/i} \tag{38}$$

where, $v_t$ represents the share of the wind storage joint venture containing a cooperative member $i$ of size $t$; $v_{t/i}$ represents the share of the wind storage joint venture that does not contain a cooperative member $i$ of size $t$. The profit distribution value of the cooperative member $i$ is as shown in Equation (32), and the specific income distribution of wind power suppliers and pumped storage power stations is shown in Equations (39)–(41):

$$r_i = \sum p_i \omega_i = \sum \frac{(t-1)!(m-t)!}{m!}(v_t - v_{t/i}) \tag{39}$$

$$R_{wp} = \frac{(X_{wpu} + X_{wp} - X_{pu})}{2} \tag{40}$$

$$R_{pu} = \frac{(X_{wpu} - X_{wp} + X_{pu})}{2} \tag{41}$$

where, $r_i$ represents the value of cooperative member $i$; $R_{wp}$ and $R_{pu}$ represent the benefits of the cooperative game between the wind power supplier and the pumped storage power station; $X_{wpu}$ represents the overall income of wind storage joint ventures; $X_{wp}$ represents the income of wind power suppliers' independent participation in the distribution market; $X_{pu}$ represents the income of pumped storage power stations independently participating in the distribution market.

## 4. Model Calculation and Solution

### 4.1. Interactive Planning Method

It can be seen from the above model that the model solution is solved for multi-objective nonlinear programming problems. The interactive planning method can develop multi-objective linear programming problems according to the research content of this paper, providing satisfactory solutions for participants [21]. The specific steps of the interactive planning method are as follows:

(1) Assume that the weight vector reduction factor $a$ ($0 < a < 1$) is proposed by the decision-maker, discriminating $m$ ($1 \leq m \leq 2t+1$) sample points, $S$ is the number of iterations, and $\varphi$ is a small positive number.

(2) Assume $q = 1 - a^{1/(t-1)}$, $i_0 = 0$, $h_i^{i_0 1} = \infty$, $i = 1, 2, \cdots, m$, $s = 1$, $G^{k1} = (\varphi, \cdots, 1, \cdots, \varphi)^T$, $k = 1, 2 \cdots, t$, then $G^{t+1,s} = \frac{1}{t} \sum\limits_{k=1}^{t} G^{ks}$, $G^{t+1+i,s} = \frac{1}{t}[\sum\limits_{k \neq i} G^{kt} + G^{t+1,s}]$, $i = 1, 2, \cdots, m$.

(3) Obtain that $q(G^{ks})(k = 1, L, 2t+1)$. Optimal target function value is $f^{ks} = f(x^{ks})(k \neq i_{s-1})$. From $\left\{ f^{1s} \quad \cdots \quad f^{2t+1,s} \right\}$ choose $f^{1s}, \cdots, f^{js}, f^{i_{s-1}s}$ as different from each other. If $j \leq m - 1$, turn (4), Otherwise, use the one-dimensional screening method to screen out $m - 1$ $f^{p_1 s}, \cdots, f^{p_{m-1},s}$ as the most different.

(4) Determine if there is a satisfactory solution for the participants in $\left\{ f^{p_1 s}, \cdots, f^{p_{m-1},s}, f^{i_{s-1}s} \right\}$. If it exists, recorded as $f^{is}$, and turn (5).

(5) If there is $f^{is}$ or $s = S$, getting the optimal solution $x^{i,s}$, and end of solution.

### 4.2. Model Specific Solution Process

Based on the above interaction planning method, the process of solving the double-layer optimization problem of the wind storage joint venture participating in the power market is shown in Figure 2.

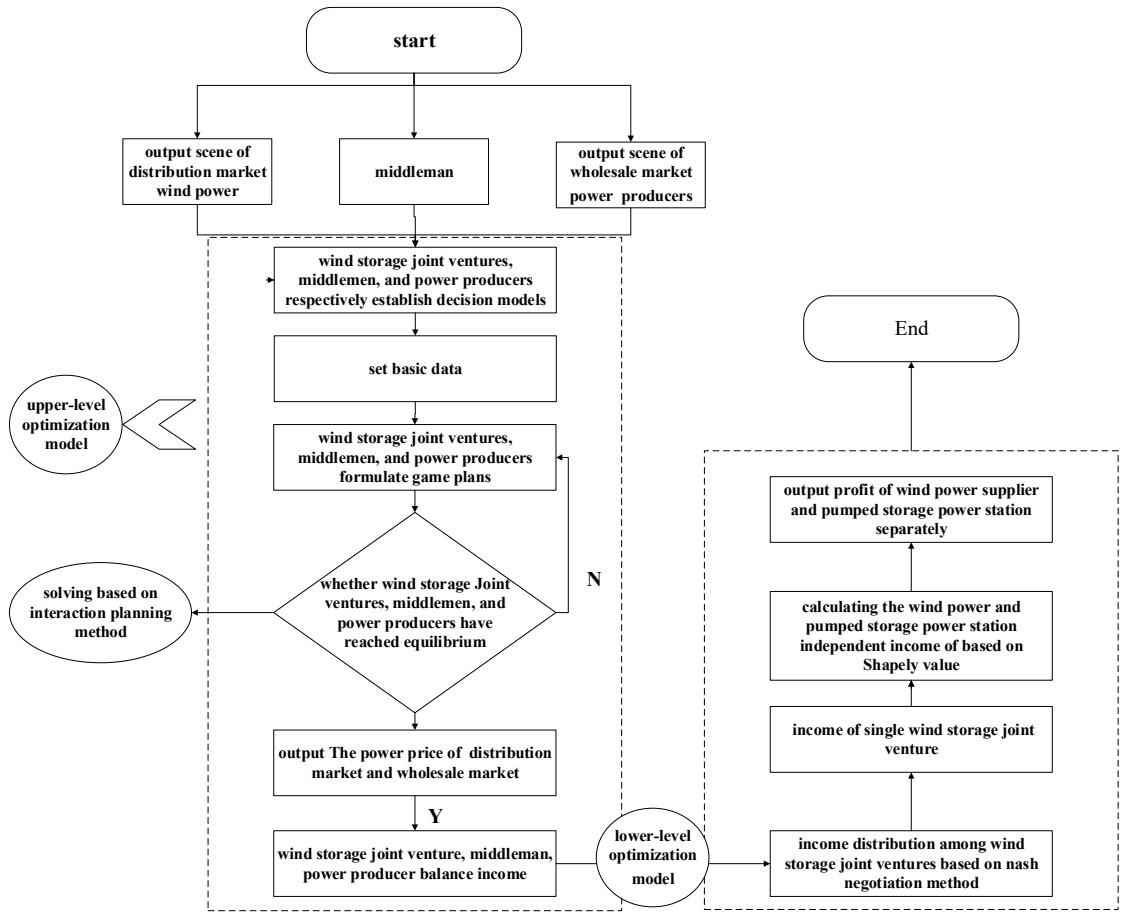

**Figure 2.** Model specific calculation flow chart.

The specific solution steps are as follows:

Step 1: Set the basic data including cost coefficient, market demand, and bidding strategy parameters of each market entity.

Step 2: Determine the equilibrium state. Through the interactive planning method, it is judged whether there is a solution that satisfies the generator, the middleman, and the wind storage joint venture at the same time, and whether the three subjects are in equilibrium at the same time. If the three subjects reach the equilibrium state at the same time, output the equilibrium game plan of the three subjects and the distribution market and the wholesale market's equilibrium power price. If the equilibrium state is not reached, the wind storage joint venture, the middleman, and the pumped storage power station adjust their respective game plans until they reach the equilibrium state at the same time.

Step 3: Calculate the profit of the three entities. Based on the respective equalization game schemes and objective functions of the three subjects in step 2, the respective profits are solved.

Step 4: Determine the distribution matrix. Based on the distribution plan proposed by all wind storage joint ventures, the overall distribution matrix is formed, obtaining the highest and lowest profit distribution ratio of each wind storage joint venture.

Step 5: Determine the allocation plan. The Nash negotiation method is used to obtain an allocation plan that satisfies each wind storage joint venture. The allocation ratio of each wind storage joint venture in the distribution plan must be greater than the lowest profit distribution ratio. Otherwise, negotiations will continue until the conditions are met.

Step 6: Calculate the profit of wind power suppliers and pumped storage power stations. According to the distribution plan in step 5, the profit of each wind storage joint venture is obtained in

Matlab. We can then find the profit of the wind power supplier and the pumped storage power station to participate in the competition independently, determining the profit obtained by the wind power supplier and the pumped storage power station through the Shapely value.

## 5. Results and Discussion

### 5.1. Basic Data

Assume that in the example, there are two generators in the market ($E_1$, $E_2$). There are four wind power providers in the distribution market ($W_1$, $W_2$, $W_3$, $W_4$) and two pumped storage power stations ($P_1$, $P_2$). The wind power supplier $W_1$, $W_2$ and the pumped storage power station $P_1$ form the wind storage joint venture $J_1$. The wind power supplier $W_3$, $W_4$, and the pumped storage power station $P_2$ form the wind storage joint venture $J_2$. Thus, two wind storage joint ventures ($P_1$, $P_2$) are formed in the distribution market. The demand in the wholesale market is 800 MW, and the demand in the distribution market is 300 MW. The parameter settings of power producers, wind storage joint ventures, and middlemen are shown in Table 1 [22,23].

**Table 1.** Related parameters of power producers, wind storage joint ventures, and middlemen.

| | Variable Cost Factor $a$ | Variable Cost Factor $b$ | Fixed Cost Factor $c$ | Upper Limit of Bid Strategy | Upper Limit of Bid Strategy |
|---|---|---|---|---|---|
| power producer $E_1$ | 0.02 | 6 | 4 | 2.5 | 0 |
| power producer $E_2$ | 0.015 | 5 | 3 | 3.5 | 0 |
| Middleman $M$ | — | — | — | 0.3 | 0 |
| wind storage joint venture $J_1$ | 0.01 | 4 | 2 | 4.2 | 0 |
| wind storage joint venture $J_2$ | 0.005 | 3 | 1 | 8.7 | 0 |

For convenience of calculation, assume that the upper limit of the wind power supplier's output is 200 MW. The parameter settings of the pumped storage power station are shown in Table 2 [24].

**Table 2.** Relevant parameters of pumped storage power station.

| | $\lambda$ | $S_{min}^{upper}$ (MW·h) | $S_{max}^{upper}$ (MW·h) | $S_{min}^{under}$ (MW·h) | $S_{max}^{under}$ (MW·h) | $Q_{max}^{ch}$ (MW·h) | $Q_{min}^{ch}$ (MW·h) | $Q_{max}^{dis}$ (MW·h) |
|---|---|---|---|---|---|---|---|---|
| pumped storage power station $P_1$ | 0.75 | 10 | 180 | 0 | 180 | 6 | 0 | 6 |
| pumped storage power station $P_2$ | 0.80 | 20 | 200 | 0 | 200 | 8 | 0 | 8 |

### 5.2. Empirical Analysis

According to the above parameters and the decision model of the power producers, middlemen, and wind storage joint ventures, the equilibrium results of the upper and lower models are shown in Table 3.

**Table 3.** Upper and lower model equilibrium results.

|           | Boost Amount (MW) | Power Price ($/MW·h) | Cost ($/h) | Profit ($/h) |
|-----------|-------------------|----------------------|------------|--------------|
| $E_1$     | 365.20            | 3.84                 | 686.60     | 716.00       |
| $E_2$     | 457.60            | 3.84                 | 766.99     | 990.48       |
| $M$       | 22.80             | 3.84                 | 87.57      | 10.62        |
| $J_1 + J_2$ | 277.20          | 4.31                 | 168.77     | 1025.02      |
| $J_1$     | 102.90            | 4.31                 | 73.35      | 374.16       |
| $J_2$     | 174.30            | 4.31                 | 88.85      | 662.60       |
| $P_1$     | —                 | —                    | —          | 122.64       |
| $P_2$     | —                 | —                    | —          | 168.30       |

As can be seen from Table 3, the wholesale market power price of 3.84 $/MW·h is lower than the distribution market power price of 4.31 $/MW·h. The middleman bids 22.8 MW of electricity from the wholesale market at the price of 3.84 $/MW·h, sold at the distribution market for 4.31 $/MW·h, earning a profit of 10.62 $/h. In the distribution market, the total bidding amount of wind water storage joint venture is 277.20 MW, and the total profit is 1025.02 $/h. Based on the lower-level profit distribution model, the profit of the wind storage joint venture and the pumped storage power station can be obtained, respectively.

5.2.1. The Impact of Different Alliance Scenarios on Equilibrium Results

To verify that wind power suppliers and pumped storage participate in the power distribution market in a cooperative manner (surpassing other methods), three different alliance scenarios are set up to study their impact on the equilibrium results. The scenario settings are as follows:

Scenario 1: The wind power supplier alliance, pumping storage power stations participate in competition independently.

Scenario 2: The wind power suppliers independently participate in competition, pumped storage power stations alliance.

Scenario 3: The wind power suppliers and the pumped storage power stations form a wind storage joint venture to participate in the competition.

The boost amount of each subject in different scenarios is shown in Figure 3. The total number of bids for wind power suppliers and pumped storage power stations is the sum of the scalar quantities of wind power suppliers and pumped storage power stations. Pumped storage power stations can be bought and stored when the market power price is low, and sold when the market power price increases (to obtain profits).

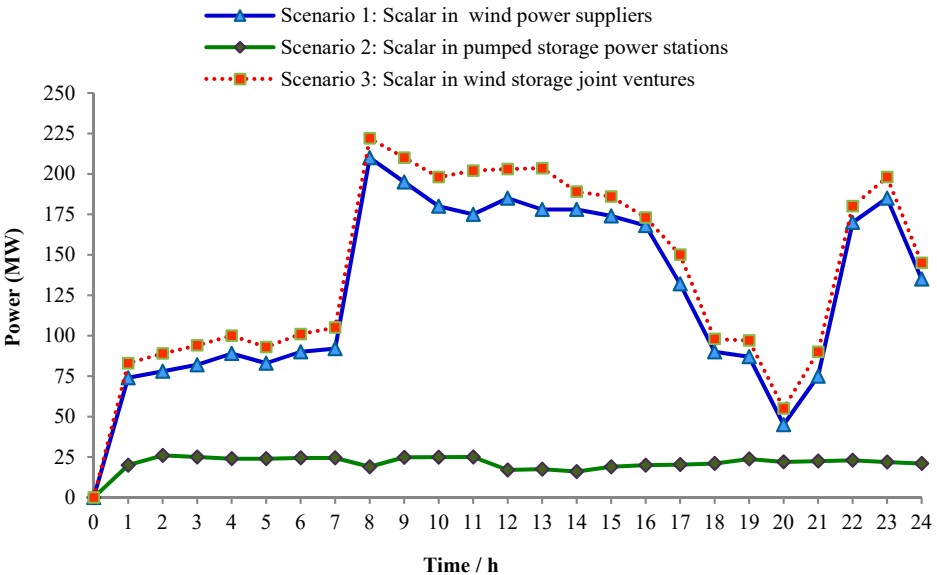

**Figure 3.** Scalar in each subject.

It can be seen from Figure 3 that the wind storage joint venture's boost amount is higher than those of the wind power supplier alliance and the pumped storage power station alliance. This is because only the wind power supplier alliance is subject to the volatility of its output and the wind power business decision-making behavior is more conservative to avoid risks. However, after the wind power supplier and the pumped storage power station are combined, the pumped storage can flexibly adjust its output force deviation, buffering the risk inside the wind storage joint venture, so the bid amount in the market will increase accordingly. The specific income of each subject under different alliances is shown in Table 4. It can be seen that the income of the wind storage alliance is higher than those of the pumped-storage power station alliance and the wind power supplier alliance. This is because the distribution market's power price is fixed, and with the increase of the boost amount, the profits of wind power suppliers and pumped storage power stations will increase accordingly. Furthermore, the 19.3% revenue growth rate of pumped storage power stations is lower than that of the wind power supplier (23.38%). Because the wind power supplier and the pumped storage power station alliance has a great influence on the decision-making behavior of the wind power supplier, the change of wind power supplier declaration rate is greater than that of the pumped storage power station, resulting in a higher income growth rate.

**Table 4.** Profit of different alliances.

| \multicolumn{6}{c}{Alliance Situation} | | | | | | \multicolumn{2}{c}{Profit (\$)} | | \multicolumn{2}{c}{Growth Rate of Profit} | |
|---|---|---|---|---|---|---|---|---|---|
| $W_1$ | $W_2$ | $W_3$ | $W_4$ | $P_1$ | $P_2$ | Wind power supplier | Pumped storage power station | Wind power supplier | Pumped storage power station |
| Y | Y | Y | Y | N | N | 604.51 | — | — | — |
| N | N | N | N | Y | Y | — | 234.79 | — | — |
| Y | Y | Y | Y | Y | Y | 745.81 | 290.95 | 23.28% | 19.30% |

Note: *Y* indicates that the subject is a member of the alliance, *N* indicates that the subject is not a member of the alliance.

5.2.2. Operation Efficiency of Pumped Storage Power Station and the Influence of Presence or Absence of Middlemen on Equilibrium Results

The decision-making behavior of wind-storage joint ventures participating in the power distribution market competition is affected by many factors, especially the operating efficiency of pumped storage power stations. Therefore, studying the effect of P1 on the equilibrium results when

the operating efficiency is 0.70, 0.75, and 0.80, but the other parameters are unchanged. The specific results are shown in Figures 4 and 5.

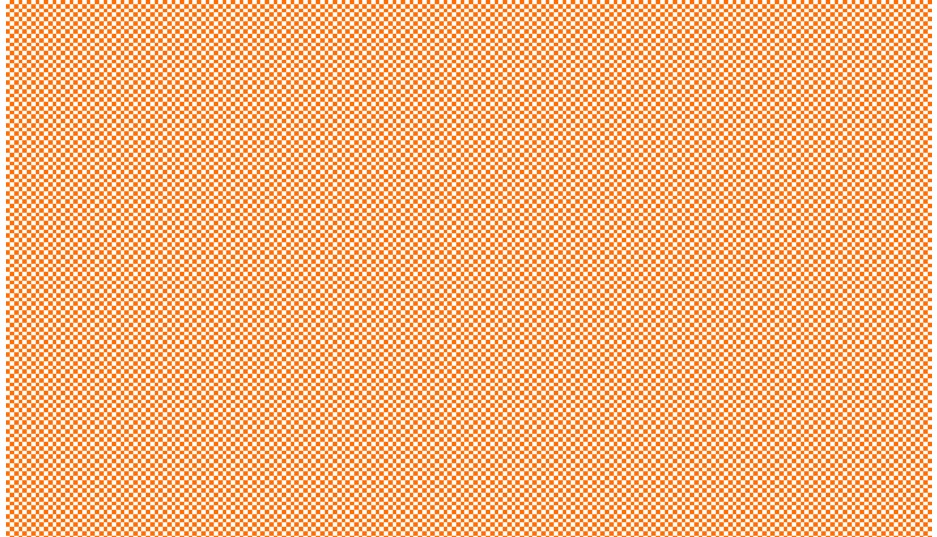

**Figure 4.** Profit chart of each entity.

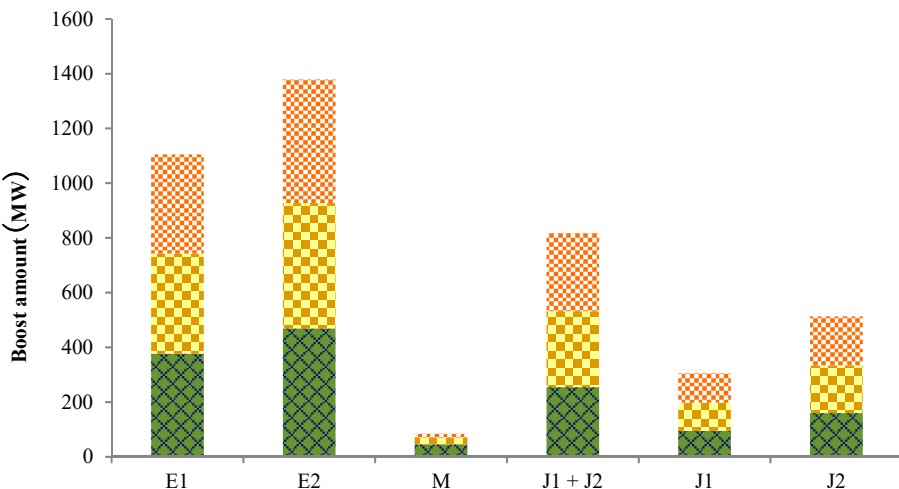

**Figure 5.** Boost amount of each entity.

It can be seen from Figures 4 and 5 that the operating efficiency of the pumped storage power station increases from 0.7 to 0.8, and the scalar quantity and profit of the wind storage joint venture will increase accordingly. The operation efficiency of the pumped storage power station will reduce the operating cost of the unit, improve the flexibility of adjustment, and enable the wind storage joint ventures to improve the decision parameters and boost amount in the market, ultimately increasing the profit of wind storage joint ventures. However, the increased supply of wind storage for the joint ventures will lead to a decline in the distribution market's equilibrium electricity price. The decrease in the power price difference between the distribution market and the wholesale market will cause the middlemen to lower their decision-making behavior parameters. As a result, the denominator's declared power is reduced, which makes the profit of the middlemen decrease with the increase of the pumped storage unit's operating efficiency. The decrease of the middlemen's boost amount in the market will cause the decline of the market's demand and will cause the wholesale market's power price and the power producers' boost amount to decline. The simultaneous changes in income and cost leads to a large drop, then a slight rise.

To study the mechanism in the distribution and wholesale markets, we established two scenarios (with and without middlemen) to analyze the impact of the equilibrium results. The changes of power prices in the two markets (with and without middlemen) are shown in Figure 6. Here, *a*, *b*, *c*, *d*, *e*, *f*, *g*, *h*, *i*, *j* represent the power price of the distribution market with middlemen; the power price of the wholesale market with middlemen; the power price of the distribution market without middlemen; the power price of the wholesale market without middlemen; two market power price difference with middlemen; two market power price difference without middlemen; power price fluctuations in the distribution market with middlemen; power price fluctuations in the wholesale market with middlemen; power price fluctuations in the distribution market without middlemen; and power price fluctuations in the wholesale market without middlemen, respectively.

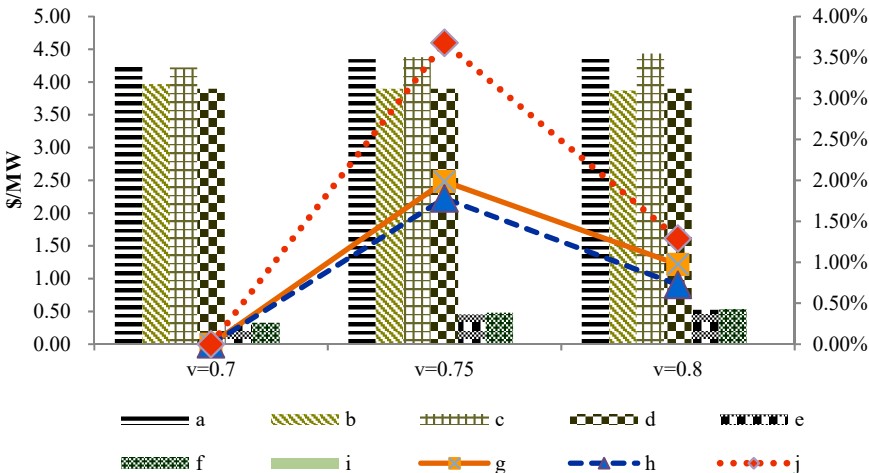

**Figure 6.** Power price for each market with or without middlemen.

It can be seen from Figure 6 that when there is no middleman, the operating efficiency of the pumped storage power station increases from 0.70 to 0.85, and the power price in the distribution market rises rapidly. The rate of increase is 1.99% and 0.97%, respectively, but the power price of the wholesale market remains unchanged. When there is a middleman, the wholesale market's power price drops, but the rate of power price change in the distribution market is 1.78% and 0.72% which is flatter than with the absence of middlemen. The middlemen will adjust the decision-making behavior according to the power price difference between the two markets, and change the demand of the two markets to smooth the fluctuation of the market power price, thus ensuring the smooth operation of the power market.

### 5.2.3. The Impact of Wholesale Market Demand on Equilibrium Results

The formation of equilibrium between the wholesale market and the distribution market depends on the relationship between the supply and the demand in the two markets. Therefore, taking the wholesale market demand change as an example, we assume the distribution market's demand is 300 MW in a certain hour period. When the wholesale market demand is 700, 750, 800, 850, and 900 MW, the impact on the equilibrium of the two markets equilibrium results is shown in Table 5.

**Table 5.** Impact of wholesale market demand on the equilibrium results.

| Demand of Wholesale Market | Market Entity | Boost Amount (MW) | Power Price ($/MW·h) | Cost ($/h) | Profits ($/h) |
|---|---|---|---|---|---|
| 700 MW | $E_1$ | 365.80 | 3.78 | 688.35 | 695.90 |
| | $E_2$ | 458.60 | 3.78 | 769.64 | 965.78 |
| | $M$ | 24.40 | 3.78 | 92.33 | 11.82 |
| | $J_1 + J_2$ | 275.60 | 4.27 | 167.57 | 1008.82 |
| 750 MW | $E_1$ | 365.20 | 3.84 | 686.60 | 716.00 |
| | $E_2$ | 457.60 | 3.84 | 766.99 | 990.48 |
| | $M$ | 22.80 | 3.84 | 87.57 | 10.62 |
| | $J_1 + J_2$ | 277.20 | 4.31 | 168.77 | 1025.01 |
| 800 MW | $E_1$ | 364.40 | 3.94 | 684.28 | 752.30 |
| | $E_2$ | 456.20 | 3.94 | 766.99 | 1037.01 |
| | $M$ | 20.60 | 3.94 | 81.21 | 9.05 |
| | $J_1 + J_2$ | 279.40 | 4.38 | 170.38 | 1053.79 |
| 850 MW | $E_1$ | 363.70 | 3.98 | 682.24 | 764.92 |
| | $E_2$ | 455.10 | 3.98 | 760.40 | 1050.46 |
| | $M$ | 18.80 | 3.98 | 74.80 | 8.07 |
| | $J_1 + J_2$ | 281.20 | 4.41 | 171.74 | 1067.87 |
| 900 MW | $E_1$ | 363.10 | 4.00 | 680.50 | 772.48 |
| | $E_2$ | 454.40 | 4.00 | 758.55 | 1059.78 |
| | $M$ | 17.50 | 4.00 | 70.03 | 7.29 |
| | $J_1 + J_2$ | 282.50 | 4.42 | 171.74 | 1070.64 |

The demand of the wholesale market increased from 700 MW to 900 MW, and the power producers have increased the decision-making parameters. The market's boost amount has increased, and the equilibrium power price formed by the wholesale market has risen correspondingly, resulting in the power price decrease between the wholesale market and the distribution market. The decrease in power price difference has prompted middlemen to reduce their decision-making parameters. The amount of electricity purchased from the market has decreased, that is, the amount of electricity sold to the distribution market has decreased. The corresponding boost amount of the wind storage joint ventures increased to maintain the market balance between supply and demand. The profit of the middlemen decreased from 11.82 $/h to 7.29 $/h eventually, but the total profit of the wind storage joint venture increased from 1008.82 $/h to 1070.64 $/h.

When the demand changes the wholesale market, the profit rate of the two market entities and the bidding demand elasticity are shown in Table 6. The bidding demand elasticity refers to the percentage change of the market entity's boost amount caused by the change in the wholesale market demand by 1% in a certain period of time. It can be seen from Table 6 that the bidding demand elasticity of middlemen is negative and the elasticity value is too large, indicating that the market demand change has a significant impact on the boost amount of the middlemen. It also can be seen from the profit rate change of each entity that the change in market demand has a greater impact on the profit of the middlemen.

**Table 6.** Change in profit rate and bidding demand elasticity.

| | Demand of Wholesale Market | | | | |
|---|---|---|---|---|---|
| | **700 MW** | **750 MW** | **800 MW** | **850 MW** | **900 MW** |
| Profit rate change of $E_1$ | 0.00 | 0.03 | 0.05 | 0.02 | 0.01 |
| Profit rate change of $E_2$ | 0.00 | 0.03 | 0.05 | 0.01 | 0.01 |
| Profit rate change of $M$ | 0.00 | −0.10 | −0.15 | −0.11 | −0.10 |
| Profit rate change of $J_1 + J_2$ | 0.00 | 0.02 | 0.03 | 0.01 | 0.00 |
| Bidding demand elasticity of $E_1$ | 0.00 | −0.02 | −0.03 | −0.03 | −0.02 |
| Bidding demand elasticity of $E_2$ | 0.00 | −0.03 | −0.05 | −0.04 | −0.02 |
| Bidding demand elasticity of $M$ | 0.00 | −1.05 | −1.60 | −1.44 | −1.11 |
| Bidding demand elasticity of $J_1 + J_2$ | 0.00 | 0.09 | 0.12 | 0.10 | 0.07 |

## 6. Conclusions

To stabilize the bidding bias caused by the volatility of wind power suppliers, this paper proposes that wind power suppliers and pumped storage power stations form an alliance to participate in the power market competition. The middlemen were introduced to construct a two-layer stochastic optimization model for bidding competition between power producers, middlemen, and wind storage conglomerates in the market and distribution market. Researchers have not yet designed the participation of wind storage joint ventures in the power market competition mechanism and its income distribution. Therefore, the two-layer stochastic optimization model constructed in this paper is of great significance, and can provide reference for the domestic design of a subsequent competition mechanism and new energy consumption. The upper model solves the game problem between power producers, middlemen, and wind storage joint ventures. The lower model effectively distributes the profits of wind storage joint ventures through Nash value and Shapely value. The results of the example analysis indicate the following:

(1) The wind power supplier and the pumped storage power station form a wind storage joint venture to participate in the power market competition, and revenue is higher than when only the wind power supplier alliance or pumped storage power station alliance participate in the competition. As wind storage joint ventures can stabilize the volatility of wind power generation and improve the decision parameters of wind power suppliers, the overall declaration volume of wind storage joint ventures is greater than the declaration volume of wind power suppliers and pumped storage.

(2) The improved operating efficiency of pumped storage units can reduce the cost of wind storage joint ventures and increase their profits, but it will have a negative impact on the profit of middlemen (reducing their profit). Therefore, to maximize the profit of the wind storage joint venture, it is necessary to select a unit with higher operating efficiency that will improve the adjustment flexibility, reduce the loss cost, and increase the overall profit of the wind storage joint venture.

(3) The introduction of middlemen in the wholesale and distribution markets has played an important role. The existence of middlemen can ease the fluctuation of electricity prices in the distribution and wholesale markets, ensuring stable operation of the power market. The mechanism for the middlemen to ease the fluctuation of power prices is to influence the demand of the wholesale and distribution markets through the amount of declarations of the middlemen, adjusting the fluctuation of the power prices by the demand change of the two markets.

**Author Contributions:** B.M.'s contribution is data curation and writing-review& editing; S.G.'s contribution is formal analysis and writing-review& editing; C.T.'s contribution is methodology and writing-review& editing; D.N.'s contribution is Resources and supervision; Z.H.'s contribution is writing-original draft.

**Funding:** This research was funded by [the 2018 Key Projects of Philosophy and Social Sciences Research, Ministry of Education, China] grant number [18JZD032].

**Acknowledgments:** The completion of this paper has been helped by many teachers and classmates. We would like to express our gratitude to them for their help and guidance.

**Conflicts of Interest:** We confirm that the authors declare no conflict of interest.

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
