# Peer review of "Game Analysis of Wind Storage Joint Ventures Participation in Power Market Based on a Double-Layer Stochastic Optimization Model"

_processes, doi:10.3390/pr7120896_

Round 1

Reviewer 1 Report

The article presented for review is very interesting. To stabilize the tendency caused by the volatility of wind energy suppliers, the Authors proposed an alliance of wind energy suppliers and a pumped storage power plant. This will
allow this alliance to participate in competition on the energy market. The
two-layer stochastic optimization model constructed by the Authors is of great
importance and can be a reference for the national project of another
competition mechanism and new energy consumption. By carrying out a competition with wind energy suppliers and a pumped storage plant, the distribution of profits between joint ventures in this field can be determined.

Comments:

Standardize the notation of literature in the text, e.g. verses 35, 50 and 234.

Verse 328 is Step 5 – Shouldn’t be Step 6?

No reference to literatures [22] to [24].

Author Response

Point 1: Standardize the notation of literature in the text, e.g. verses 35, 50 and 234.

Response 1: We have modified the full text reference and marked it with yellow.

Point 2: Verse 328 is Step 5-Shouldn't be Step 6?

Response 2: Verse 328 is Step 6, and we have modified it and marked with yellow.

Point 3: No reference to literatures [22] to [24].

Response 3: Literatures [22] to [24] is the source of data for case analysis, and they have been supplemented.

Reviewer 2 Report

The paper presents the problem of a wind storage joint ventures participation in power market. Authors made attempts to find the equilibrium state between wind storage joint ventures, middlemen, and power producers activities. The authors have proposed a new method which described analyzed problem by means of a stochastic optimization model. The most interesting aspect of the paper is proposition of two-level distribution model of the problem. New solutions in the proposed method are state Nash negotiation method and Sheply value utilization in the optimization process.

The paper is well organized and include all important elements: discussion of existing solutions, presentation of the paper innovations, detailed description of proposed method, the method utilization results and their discussion. However, in my opinion there is a lack of justification for value selection and information about data sources.

In my opinion the paper can be accepted and published in the present form. However, some minor remarks are required - I noticed that some of the references position (22, 23 and 24) are not included in the paper.

Author Response

Point 1: However, in my opinion there is a lack of justification for value selection and information about data source.

Response: Literature [22] to [24] is the source of data for case analysis, and they have been supplemented.

Point 2: I noticed that some of the references position (22, 23 and 24)are not included in the paper.

Response: Literature [22] to [24] has been supplemented in the text.